# Impact of COVID-19 Pandemic on Caregivers of People with an Intellectual Disability, in Comparison to Carers of Those with Other Disabilities and with Mental Health Issues: A Multicountry Study

**DOI:** 10.3390/ijerph20043256

**Published:** 2023-02-13

**Authors:** Andrew Wormald, Eimear McGlinchey, Maureen D’Eath, Iracema Leroi, Brian Lawlor, Philip McCallion, Mary McCarron, Roger O’Sullivan, Yaohua Chen

**Affiliations:** 1Trinity Centre for Ageing and Intellectual Disability, Trinity College Dublin, D02 PN40 Dublin, Ireland; 2The Global Brain Health Institute, Trinity College Dublin, D02 PN40 Dublin, Ireland; 3College of Public Health, Temple University, Philadelphia, PA 19122, USA; 4Institute of Public Health, D08 NH90 Dublin, Ireland; 5The Bamford Centre, Ulster University, Coleraine BT52 1SA, UK; 6Department of Gerontology, Lille University Hospital, 59000 Lille, France; 7INSERM UMR-S 1172, Vascular and Degenerative Cognitive Disorders, University of Lille, 59000 Lille, France

**Keywords:** carers, intellectual disability, loneliness, isolation, burden, COVID-19

## Abstract

Carers supporting people with an intellectual disability often rely on others to manage the burden of care. This research aims to compare the differences between carer groups and understand the predictors of loneliness changes and burden for carers of people with an intellectual disability. Data from the international CLIC study were analysed. In total, 3930 carers responded from four groups; people who care for those with mental health difficulties (*n* = 491), dementia (*n* = 1888), physical disabilities (*n* = 1147), and Intellectual disabilities (*n* = 404). Cross tabulation and the chi-squared test were used to compare group compositions and binary logistic regression to model predictors within the intellectual disability group. A total of 65% of those caring for people with an intellectual disability experienced increased burden, and 35% of carers of people with an intellectual disability and another condition experienced more severe loneliness. Becoming severely lonely was predicted by feeling burdened by caring (AOR, 15.89) and worsening mental health (AOR, 2.13) Feeling burden was predicted by being aged between 35 and 44 (AOR, 4.24), poor mental health (AOR, 3.51), and feelings of severe loneliness prior to the pandemic (AOR, 2.45). These findings demonstrate that those who were already struggling with caring experienced the greatest difficulties during the COVID-19 lockdowns.

## 1. Introduction

Compared to pre-pandemic levels, carers of people with physical and brain health conditions experienced a significant increase in burden, loneliness, and mental health difficulties [1]. Loneliness is a subjective sense of inadequate quantity or quality of social contact [2]. Carer burden is a subjective multifaceted construct for carers including social and psychological constraints, personal strain, interference with personal life, concerns about the future, and guilt, all of which have been significantly impacted by the COVID-19 pandemic [1].

Family carers are ‘key care partners’ to formalised services, providing informal and unpaid caring to a dependent relative, and form the backbone of social care provision [3]. Caring is a multidimensional experience and carers may derive positive benefits from providing care and, simultaneously, be vulnerable to negative physical, psychological, social, and financial impacts of caregiving [4,5,6,7]. Social isolation, loneliness, and decreased social activity can increase carer burden, which is associated with increased morbidity and mortality [8,9,10,11]. 

The experience of providing informal care for a person with intellectual disability has been reported to mirror the joys, benefits, and challenges experienced by carers in other contexts [12]. However, a number of features distinguishes these carers including the longevity of the caring relationship [13,14], the impact of ageing on both the carer and the care recipient, and concerns about the future of the care recipient when the carer dies or is no longer in a position to continue caring [15,16,17,18]. Informal or family carers are also more likely to engage in intensive caring due to the prevalence of comorbid health issues often experienced by people ageing with intellectual disability [19]. 

The vulnerability of informal carers was thrown into sharp relief by the COVID-19 pandemic. State responses to the pandemic included the abrupt closure of health and social services, and restrictions on movement and social interactions, meaning carers were left with reduced supports [20,21,22,23]. In the absence of formal supports, the responsibility for providing care to children, older people, and people with an intellectual disability fell to family carers. The family home, according to Daly [24] was reaffirmed as the premier site of caring, and informal care proved to be more resilient than formal care in the context of older people [25]. Family carers of people living with intellectual disability are reliant on formal supports, particularly day and respite services, to sustain their ability to care [12], and may be particularly impacted by the abrupt closure of day, respite, and therapeutic services. In the absence of formal services, the responsibilities and duties of informal carers increased and intensified [9,26], and few of the extensive social protection response measures implemented by many countries were aimed at family carers [27]. Family carers also experienced the loss of employment with resulting financial insecurity, or the relocation of the workplace into the family home [28]. Many carers, consequently, experienced challenges reconciling the care and paid employment components of their lives [29]. Support networks for carers became compromised due to social distancing requirements and informal carer status has been identified as an independent risk factor for increased loneliness during the pandemic [1]. 

Research on experiences during the COVID-19 pandemic have consistently reported that informal carers experienced isolation, loneliness, and declines in physical and mental health and wellbeing [1,3,30,31]. Doody and Keenan’s [32] scoping review reported that, during the pandemic, people living with intellectual disability and their carers were particularly vulnerable to negative physical, social, and psychological impacts. Family carers experienced extreme anxiety about the possibility and consequences of their family member becoming infected [33] and reported feelings of hopelessness [34], abandonment, mental health problems, severe anxiety, and major depression [35].

International data from the Coping with Loneliness, Isolation and COVID-19 (CLIC) study which aimed to examine the overall psychological impact of the COVID-19 pandemic, regardless of stage of the epidemic, through validated self-report measures of loneliness and social isolation. The CLIC study received responses from over 100 countries across 10 languages, and reported significant rises in severe loneliness and isolation among carers who were already vulnerable with mental health or financial difficulties [36]. 

To date no research has considered the global experience of caring in relation to loneliness, isolation, and burden for carers of people with an intellectual disability. This research aims to examine the impact of the COVID-19 pandemic on carers of people with an intellectual disability, in comparison to carers of those with other disabilities and with mental health issues using validated self-report measures of loneliness and social isolation.

## 2. Materials and Methods

### 2.1. Study Design

CLIC was an international online survey (https://publichealth.ie/clic/, accessed on 5 February 2023) with 20,000 participants across 100 countries examining the impact of the COVID-19 pandemic on loneliness and social isolation [36]. The survey was informed by results from a preliminary study conducted by the Alzheimer’s Association of Ireland at the beginning of the COVID-19 pandemic, which suggested that careers of people living with dementia were experiencing high levels of burden and isolation [37]. 

Embedded within this survey were questions specific to carer experiences, and included carers of people with dementia, intellectual disability, physical health problems, and mental health problems. Carers were identified through the question “*Do you provide care and support to a family member or friend with a long-term or life-limiting health problem or disability (including mental health)?*”.

### 2.2. Participants

Participants were recruited through voluntary sector organisations, the charitable sector, social media, and through email lists of international organisations such as the International Association of the Scientific Study of Intellectual and Developmental Disability (IASSIDD). Participants in the survey had to be 18+ years, provide informed consent, and be able to use the internet in order to participate. Data collection took place between 2 June and 16 November 2020.

### 2.3. Research Tools

#### 2.3.1. Sociodemographic

The survey included questions on gender, age, and questions on the condition of the care recipient and caregiving circumstances. Other questions included physical and mental health of carer, measured by a 5-point Likert scale (“*Would you say that, in general, your physical health/mental health is...*”. Responses to these questions were binary coded for analysis as excellent, very good, and good (1), and fair and poor (0). Participants were also asked about their financial circumstances “*How well do you feel your needs are met by the financial resources you have (i.e., money)?*” which was coded as 1. Very well, 2. Fairly well, and 3. Poorly. 

Validated measures of loneliness, social and emotional loneliness, caregiver/carer burden, and social isolation were used in the survey and are described below. 

#### 2.3.2. Loneliness

Loneliness was measured using two scales, the deJong Gierveld scale for social and emotional loneliness [38] and the 5-item UCLA loneliness scale [39].

The six-item deJong Gierveld scale included questions about overall social and emotional loneliness with statements such as, “*During COVID-19 there are many people I can trust completely*” (Social loneliness) and, “*Before COVID-19, I experienced a general sense of emptiness*” (Emotional loneliness). Each item was offered a three-point Likert scale (No, More or less, Yes).

Loneliness was also assessed by the modified 5-item UCLA loneliness scale, which has been validated in previous studies and includes items such as, ”how often do you feel in tune with the people around you?” Response options were hardly ever (0), some of the time (1), and often (2), providing an overall score between 0 and 10 with higher scores meaning higher loneliness. Questions were asked first about “*Before COVID-19*”and then “*During COVID-19*” giving pre- and during-COVID-19 loneliness scores which were categorised as scores of 0–4 denoting none/low loneliness; 5–6 denoting moderate loneliness; and 7+ severe loneliness. Cronbach’s alpha for the UCLA scale for the overall sample was 0.77 pre COVID-19 and 0.82 during COVID-19. Within those who identified as carers of people with an intellectual disability, the Cronbach’s alpha was 0.77 pre- and 0.77 during COVID-19.

Changes in participants scores pre-COVID-19 and during COVID-19 were categorised by the change in score and binary coded accordingly.

#### 2.3.3. Caregiver/Carer Burden

Participants were asked, “*During COVID-19 how often do you feel burdened in your caring role?*” with the response options ranging from Never (1) to Nearly always (5). This was taken from the Zarit Burden Interview (ZBI) [40], which has been validated in capturing caregiver burden. For the final regression, this variable was binary coded to those who reported feeling burdened quite frequently or nearly always, against those who reported rarely or sometimes feeling burdened. Participants were also asked to respond to change in level of burden during COVID-19, “*same as usual, more than usual, less than usual*”. 

#### 2.3.4. Isolation 

Isolation pre- and during COVID-19 was captured using the validated six-item Lubben Social Network Scale (LSNS-6) [41]. This scale includes questions on social support network, frequency of contact and closeness of contact using a 5-point Likert scale providing an overall score between 0 and 30. Participants with scores <12 are defined as isolated. Participants were also asked about change in social isolation during COVID-19, “*the same, more than usual, less than usual*” with scores ranging between –6 and 6, where a score of –3 or lower indicated an increase in social isolation. Validity for the pre- and during COVID-19 scales were a = 0.83 and a = 0.72, respectively. 

#### 2.3.5. Changes during COVID-19

Participants were asked about changes to their routine during the COVID-19 pandemic around eating, sleeping, physical activity, mental health, finances, and cultural activities each with three response options less/worse (1), more/better (2), about the same (3), e.g., “*During COVID-19 are you eating less food than you did before, eating less food than you did before, eating about the same*”.

Mental health was measured with the item, “*Would you say that, in general, your mental health is*:” with the responses binary coded, Excellent, Very good, Good (1), and Fair, Poor (0). 

Employment during COVID-19 was asked about with, “*Select which best applies to your current situation*” with the response options: employed but off work due to COVID-19; employed and still going to work; working from home due to COVID-19; self-employed; looking after home or family; in education or training; unemployed because of COVID-19; unemployed; furloughed/COVID-19 employment support payment; retired; other. Responses were binary coded into working (1) and not working (0). 

### 2.4. Ethical Approval 

The study was conducted according to the guidelines of the Declaration of Helsinki, and approved by the Ethics Committee of Ulster University (RG3) on 15 May 2020.

### 2.5. Analysis

Data were analysed using SPSS v26.0 (SPSS Inc., Chicago, IL, USA).

First participants were compared by care recipient conditions, the options were as follows: care for a person living with an intellectual disability only (ID); care for a person living with dementia only (Dementia); care for somebody living with a mental health condition only (Mental health); care for somebody with a physical disability only (Physical disability); and care for a person living with an intellectual disability and at least one other chronic condition (Intellectual Disability multimorbid). Descriptive data were produced looking at differences in the composition groups and loneliness was calculated for before and during the COVID-19 pandemic. Chi-squared analysis was conducted to test for differences in groups.

Next, the categories of Intellectual Disability and Intellectual Disability multimorbid were collapsed together (Intellectual Disability Total). The changes in loneliness using the Intellectual Disability Total category were mapped out. Binary logistic regressions were run to investigate the predictors of becoming severely lonely, staying never lonely, and caregiver burden. Confidence intervals were calculated using bootstrapping set to 5000 cases.

## 3. Results

At closure of the survey, 23,609 respondents from 101 different countries had participated; of these, 20,398 responses were assessed as valid (i.e., completion of written informed consent). (Figure 1); of these 5236 identified as carers. Carers were from 88 different countries. Sixty-five percent were residents in five countries: the United States, 37%, United Kingdom, 13%, Ireland, 6.8%, France 5%, and Pakistan 4%. The remaining countries each contributed less than 4% to the sample. Two hundred and twenty-seven of the participants were carers of someone with an intellectual disability only, 1888 were carers of someone with dementia only, 491 were caring for someone with a mental illness only, 1147 cared for someone with a long-term physical condition only, and 177 cared for people living with an intellectual disability and at least one other condition. The intellectual disability multimorbid category comprised 131 (74.0%) people that cared for someone with an intellectual disability and one other additional condition (physical *n* = 80, dementia *n* = 16, and mental health *n* = 35), 30 (16.9%) cared for someone with intellectual disability who had two comorbid conditions (physical and dementia *n* = 12, mental health and dementia *n* = 4, mental health and physical *n* = 14), and 16 (9.0%) cared for someone with a combination of all three additional conditions. 

Table 1 below shows the key demographics across carer types. In all groups the carers were predominantly female (79.4%), with more than half (56.3%) being over the age of 55. In each group the majority of participants were married (70%), with more people caring for those with dementia (SR = 6.3) being married and fewer than expected caring for those with mental health difficulties (SR = −5.4). Religion was equally important across the groups (30.7%) and there were no significant differences in the amount of third level education received (71.6%). Those caring for people living with dementia (SR = 4.9), physical disability (SR = 2.0), and an intellectual disability (SR = −2.0) were overrepresented in the over 55 years old care group (Chi Sq = 128.394, *p* < 0.01). Participants rated their physical health as very good (81.4%) and their mental health as very good (78%). However, poor physical health was highest in carers of people in the intellectual disability multimorbid category (Chi Sq = 15.390, *p* < 0.01, SR = 2.7). Poor mental health was highest for those in the caring for intellectual disability multimorbid (SR = 2.8), and intellectual disability (SR = 2.1, Chi sq = 19.168, *p* < 0.01). Most participants agreed that their finances were at least meeting their needs fairly well (86.6%) However finances were meeting the needs poorly for carers in the intellectual disability multimorbid group (chi square = 40.669, *p* < 0.01). Overall, only 7.4% (*n* = 386) cared for a child; however, this was highest in carers of someone with an intellectual disability (SR = 14.6), intellectual disability multimorbid (SR = 19.5), and mental health (SR = 4.7, Chi Sq = 458.324, *p* < 0.01). Nearly one third of participants (34.5%) were frequently burdened by their caring role, with those caring for those living with dementia (SR = 5.1) and physical health (SR = −5.3) reporting most feelings of burden (Chi Sq = 105.734, *p* < 0.01). Over half (55.2%) felt an increased burden during COVID-19, (Chi Sq = 95.441, *p* < 0.01, Cramer’s V = 0.114) with those caring for people living with dementia (SR = 2.9) and intellectual disability (SR = 2.3) overrepresented. 

Across all groups severe loneliness increased during COVID-19 (Table 2), with only 291 participants (6.7%) reporting severe loneliness before COVID-19 and 1041 participants (24%) reporting severe loneliness during COVID-19. The DeJong Gierveld loneliness scale shows that more carers report high levels of social loneliness than emotional loneliness (Appendix A Table A1 and Table A2). Before COVID-19, carers for those living with an intellectual disability and multiple morbidities reported the largest percentage experiencing the highest levels of emotional loneliness (18.7%), during COVID-19 those caring for people living with an intellectual disability only had the highest percentage experiencing the highest levels emotional loneliness (27.4%). Social loneliness before COVID-19 was experienced by more carers for those living with mental health issues at the highest level (51.4%). During COVID-19, the most social loneliness at the highest level was reported by carers for people with an intellectual disability and multiple morbidities (66.4%).

The intellectual disability only and the intellectual disability multimorbid groups were collapsed into a single group (intellectual disability total, *n* = 404); of these 351 were caring for a family member or relative, 48 were caring for a non-relative, and 5 were unknown. For this intellectual disability total group, the trajectories of loneliness were calculated using the UCLA loneliness scale (Figure 2). The largest categorisation was not lonely; however, the numbers in this group decreased (pre *n* = 250, during *n* = 159) the most. Only six carers reported becoming not lonely during COVID-19 and 56 participants became severely lonely. Overall half of all participants (51%) reported some experience of loneliness during COVID-19.

A series of binary logistic regressions were conducted, for the intellectual disability total group only, to estimate variables that predicted moving to severe loneliness during COVID-19, never being lonely during COVID-19, and feelings of burden in the caring role. All analyses included the creation of bootstrapped confidence intervals set to 5000 cases.

The first regression was used to test which pre-COVID variables would predict moving to loneliness. Those who became severely lonely were coded 1, all others coded 0 (Table 3). Significant predictors of moving to loneliness were reporting poor mental health (AOR = 2.03, *p* < 0.05), burdened by caring role frequently (AOR = 6.66, *p* < 0.05), and burdened nearly always (AOR = 15.90, *p* < 0.01).

A second binary logistic regression was conducted to understand which variables of change during COVID-19 predicted a move to severe loneliness (Table 4). The model predicted between 20.8% and 34.3% of the move to severe loneliness. The only significant predictor was mental health worse than before (AOR = 10.19, *p* < 0.01).

A binary logistic regression was then conducted to understand which variables predicted never being lonely during COVID-19 (Table 5). The model predicted between 22.9% and 30.6% of the Never lonely variable, the significant predictors were Being Frequently burdened (AOR = 0.156, *p* < 0.01), Always burdened (AOR = 0.202, *p* < 0.05), and Excellent mental health (AOR = 3.096, *p* < 0.01)

Finally, a binary logistic regression was run to test which variables predicted feelings of burden (Table 6). The model predicted between 13.4% and 18.4% of the burden variable. When compared to the youngest age group, those aged between 35 and 69 are all significantly more burdened (AOR, 3.141 to 4.235, *p* < 0.05). Those with fair/poor mental health were twice as likely to experience burden as those with good mental health (AOR = 3.508 *p* < 0.01) and experiencing severe loneliness before COVID-19 (AOR = 2.494, *p* < 0.05).

## 4. Discussion

All groups of carers reported increases in feelings of loneliness. Particularly notable is the amount of social loneliness felt for all groups of carers during the pandemic with around nearly two-thirds reporting the highest levels of loneliness during the COVID-19 pandemic. The results show that carers of people with an intellectual disability were more likely to experience severe levels of loneliness, increased burden, and poorer mental health than people in the other categories of carers. 

For carers of people living with an intellectual disability, becoming severely lonely during the pandemic was predicted by feeling burdened frequently or always by their caring prior to service closures, those with fair or poor mental health, and those who felt their mental health had worsened during the pandemic. Those who never reported loneliness were more likely to have good mental health and less likely to have feelings of burden. Feeling burden was predicted by poor mental health, being aged between 35 and 69, with those aged 35–44 the most likely to feel burden and severe loneliness pre-pandemic. 

The results add voice to other research that has found the service closures during the pandemic had a negative impact on carers, leading to increased loneliness, declining mental health, and reduced feelings of wellbeing [9,22,31,42,43].

It is known that loneliness affects carers of people with intellectual disabilities because of the all-encompassing role and the experience of the loss of social roles [44]. It is therefore, no surprise that it is in this area where the strongest indications of loneliness lie, and where the effects of the pandemic lockdowns were most sharply felt.

Research has indicated that carers of people with an intellectual disability during this period experienced increased burden, poor social support, increased costs, and loss of employment, leading to high levels of stress and depression [35,45]. Not all research has been negative about the experience, and some carers have talked of the positives and their gratitude to those who helped them cope [34].

These results are relevant beyond COVID-19 studies as they indicate that for informal carers there is a strong relationship between poor mental health, feelings of burden, and severe loneliness. Carers for people with an intellectual disability have been found to experience relatively high levels of burden [46]. Others have demonstrated that poor mental health leads to increased levels of burden in carers [47] with burden and depression being found to have strong links in carers for people with an intellectual disability [48,49,50]. In their research, Bahtia noted that 39% of their participants experienced high levels of burden, in keeping with the pre-COVID-19 levels reported here. Furthermore, the links between loneliness and mental health difficulties are well reported in the loneliness literature. Whilst others have investigated the link between stress, loneliness, and poor mental health [51], no one has considered the linkages between poor mental health, loneliness, and burden in carers of people with an intellectual disability. Therefore, further research into the interplay between loneliness, burden, and poor mental health should be undertaken with carers of people with an intellectual disability.

### 4.1. Policy and Practice

If future pandemics hit, governments need to be cognisant of the impact of removing services from carers and those they care for. The findings document that there were significant adverse effects from the lockdown approaches taken with good intention to protect vulnerable older adults and persons with disabilities and their carers. For future epidemics and similar situations it appears that planned interventions should not only protect but support, and therefore include specific strategies for addressing respite, telesupport, and in-home needs.

Additionally, the large amounts of social loneliness felt by the carers should be acknowledged, and supports from service providers should be adapted to suit the needs of the informal carers.

### 4.2. Limitations

This study took in the voices of people from around the world during the pandemic. However, it is in the nature of psychological research to investigate those who experience problems as this can provide richer more in depth understanding about the issue under investigation.

The relatively small numbers of carers for people with an intellectual disability means the results should only be taken as indicative for any jurisdiction. Further research around the relationship between burden, loneliness, and poor mental health should be conducted. 

The sample, given the methods used, cannot be considered representative. Additionally, individuals who completed the online questionnaire were not all in the same moment of lockdowns. There is potential for sample bias, for example it may be that the voice of those who were negatively affected by the lockdowns was overrepresented.

Country of residence and ethnicity were not addressed in this study. Ethnicity could not be addressed, as in many countries, the collection of such information was not permitted. It would be useful for future research to look at the effects of ethnicity within nations. 

It is known that loneliness rates and causes vary from country to country and there may be cultural influences not accounted for in this research. However, this research was aimed at understanding if there were general effects internationally that were felt by carers of people with an intellectual disability. Future research using the same dataset should look to analyse cultural differences in responses.

## 5. Conclusions

Pandemic lockdowns removed supports from carers of all people. This research demonstrates that the impact of this removal of supports had significant impact on all carers, which was particularly severe on carers of people with an intellectual disability. Carers of people with an intellectual disability who were already struggling to cope because of feelings of loneliness, burden, and poor mental health were particularly negatively affected. Policy changes may be needed to ensure that services are not locked down in future and services should be given the scope for adaptations to meet the needs of carers.

## Figures and Tables

**Figure 1 ijerph-20-03256-f001:**
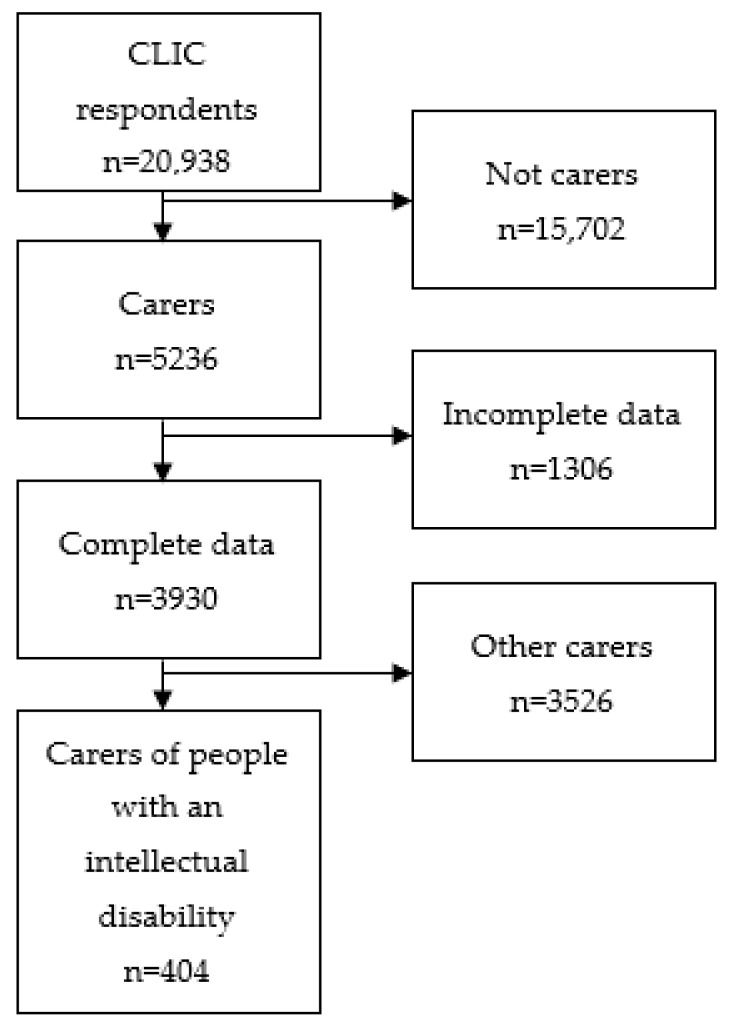
Flow diagram of participant numbers.

**Figure 2 ijerph-20-03256-f002:**
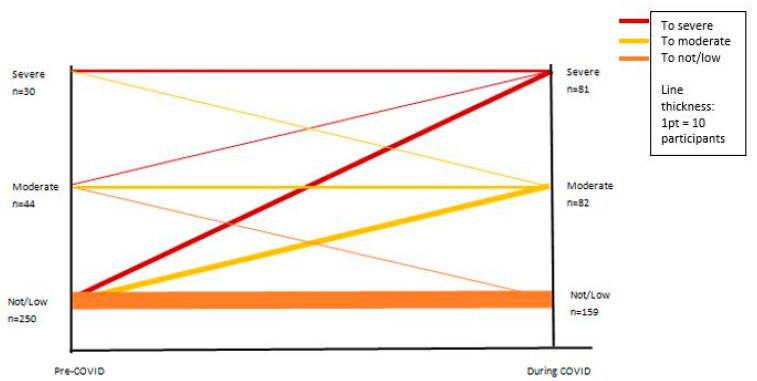
Trajectories of loneliness pre- and during COVID-19 for carers of people living with an intellectual disability (*n* = 324).

**Table 1 ijerph-20-03256-t001:** Key demographics of carers in the CLIC study.

	Intellectual Disability *n* = 227	Dementia *n* = 1888	Physical Disability *n* = 1147	Mental Health *n* = 491	Intellectual Disability Multimorbid *n* = 177
Female	82.4%	77.0%	80.4%	77.4%	80.2%
Age 55+	† 47.3%	* 67.2%	53.5%	† 41.7%	50.3%
Marital status (married/cohabiting)	67.7%	* 74.8%	67.7%	† 59.4%	65.9%
Religion very important	28.8%	30.7%	31.2%	29.7%	30.7%
Third Level education	64.4%	70.5%	73.8%	74.8%	70.4%
Poor physical health	21.6%	17.6%	16.1%	20.5%	* 27.2%
Poor mental health	* 28.4%	21.0%	19.4%	22.7%	* 32.1%
Finances meet needs poorly	* 16%	11.0%	12.4%	16.0%	* 21.7%
Care for child	* 31.3%	† 0.0%	6.3%	* 12.0%	* 24.9%
Burdened by caring role quite frequently	37.8%	* 42.2%	† 25.2%	† 26.4%	41.7%
Covid burden change more than usual	* 65.1%	* 60.1%	† 47.9%	† 44.4%	62.8%

* *p* < 0.01 and SR > 1.96, † *p* < 0.01 and SR < −1.96.

**Table 2 ijerph-20-03256-t002:** Categorised UCLA loneliness scale pre- and during COVID-19 by Carer Type.

	Intellectual Disability *n* = 175		Dementia *n* = 1644		Mental Health *n* = 404		Physical Health *n* = 988		Intellectual Disability Multimorbid *n* = 149	
	Pre	During	Pre	During	Pre	During	Pre	During	Pre	During
1: None/low	81.1%	52.9%	78.3%	48.2%	76.7%	52.7%	79.7%	54.0%	72.5%	45.3%
2: Moderate	12.6%	31.0%	14.7%	24.7%	15.1%	28.4%	15.0%	25.2%	14.8%	18.9%
3: Severe	6.3%	16.1%	7.1%	27.1%	8.2%	18.9%	5.4%	20.9%	12.8%	35.8%

**Table 3 ijerph-20-03256-t003:** Predisposing variable predictors of moving to severe loneliness (*n* = 293).

	B	AOR	Std. Error	*p*	95% Confidence Interval	
					Lower	Upper
Gender Female (reference)						
Gender Male	−0.087	0.917	0.499	0.848	−1.183	0.787
Age 18–34 (reference)						
Age 35–44	0.463	1.589	2.479	0.500	−0.897	2.505
Age 45–54	−0.772	0.462	2.652	0.294	−2.473	1.323
Age 55–69	0.008	1.008	2.457	0.974	−1.211	1.865
Age 70 and over	0.500	1.649	2.481	0.460	−0.938	2.441
Physical Health Good/Excellent (reference)						
Physical Health Fair/Poor	−0.272	0.762	0.466	0.525	−1.269	0.552
Mental Health Good/Excellent (reference)						
Mental Health Fair/Poor	0.785	2.192	0.415	0.034 *	0.007	1.626
Work Status Not Working (reference)						
Work Status Working	−0.501	0.606	0.424	0.205	−1.43	0.254
Burdened Never (reference)						
Burdened Rarely	1.482	4.401	8.775	0.084	−0.229	20.027
Burdened Sometimes	0.655	1.925	8.814	0.279	−1.109	19.183
Burdened Frequently	1.896	6.661	8.808	0.049 *	0.266	20.508
Burdened Always	2.766	15.897	8.808	0.012 **	0.967	21.648
Constant	−3.032	0.048	9.016	0.002	−21.906	−1.578

* *p* < 0.05, ** *p* < 0.01.

**Table 4 ijerph-20-03256-t004:** Precipitating variables predicting move to severe loneliness (*n* = 314).

	B	AOR	Std. Error	*p*	95% Confidence Interval	
					Lower	Upper
Food consumption About the same (reference)						
Food consumption Less than before	−0.201	0.818	0.688	0.737	−1.692	0.991
Food consumption More than before	0.107	1.112	0.416	0.787	−0.706	0.918
Sleep patterns About the same (reference)						
Sleep patterns Less than before	0.846	2.330	2.799	0.238	−1.371	2.477
Sleep patterns More than before	0.754	2.125	0.448	0.064	−0.022	1.76
Physical activity About the same (reference)						
Physical activity Less than before	0.797	2.218	0.521	0.085	−0.099	1.992
Physical activity More than before	1.054	2.869	0.729	0.071	−0.186	2.483
Mental health About the same (reference)						
Mental health Better than before	−17.921	0.000	1.817	0.999	−19.292	−15.09
Mental health Worse than before	2.322	10.192	1.607	0.000 **	1.514	4.099
Cultural activities About the same (reference)						
Cultural activities Less than before	−0.004	0.996	0.419	0.989	−0.868	0.785
Cultural activities More than before	−0.319	0.727	0.697	0.530	−1.629	0.685
Constant	−4.487	0.011	1.654	0.000	−6.698	−3.575

** *p* < 0.01.

**Table 5 ijerph-20-03256-t005:** Predictors of never lonely (*n* = 290).

	B	AOR	Std. Error	*p*	95% Confidence Interval	
					Lower	Upper
Gender Female (reference)						
Gender Male	0.329	1.39	0.414	0.393	−0.456	1.163
Age 18 to 34 (reference)						
Age 35 to 44	0.027	1.027	0.665	0.962	−1.218	1.391
Age 45 to 54	1.032	2.806	0.612	0.060	−0.02	2.377
Age 55 to 69	0.986	2.681	0.583	0.056	−0.005	2.276
Age 70+	0.445	1.561	0.691	0.483	−0.82	1.901
Never Burdened (reference)						
Rarely Burdened	−0.437	0.646	0.821	0.467	−1.87	0.76
Sometimes Burdened	−1.097	0.334	0.77	0.029	−2.489	−0.117
Frequently Burdened	−1.856	0.156	0.803	0.001 **	−3.351	−0.849
Always Burdened	−1.602	0.202	1.081	0.015 *	−3.385	−0.33
Poor Physical Health (reference)						
Excellent Physical Health	0.541	1.718	0.44	0.182	−0.266	1.465
Poor Mental Health (reference)						
Excellent Mental Health	1.13	3.096	0.437	0.004 **	0.409	2.111
Isolated (reference)						
Not Isolated	0.566	1.761	0.389	0.116	−0.178	1.353
Constant	−1.443	0.236	0.997	0.054	−3.238	0.163

* *p* < 0.05, ** *p* < 0.01.

**Table 6 ijerph-20-03256-t006:** Predictors of burden (*n* = 265).

		AOR	Std. Error	*p*	95% Confidence Interval	
					Lower	Upper
Gender Male (reference)						
Gender Female	0.630	1.878	0.438	0.114	−0.141	1.597
Age: 18–34 (reference)						
Age: 35–44	1.443	4.235	1.553	0.012 *	0.261	3.157
Age: 45–54	1.420	4.136	1.539	0.010 *	0.333	3.138
Age 55–69	1.145	3.141	1.523	0.036 *	0.117	2.852
Age: 70 and over	0.739	2.094	1.566	0.279	−0.599	2.471
Mental Health good/excellent						
Mental Health fair/poor	1.255	3.508	0.349	0.000 **	0.66	2.013
Lubben Social Network: Not Isolated (reference)						
Lubben Social Network: Isolated	−0.206	0.814	0.368	0.549	−0.959	0.468
UCLA pre-COVID-19: Low						
UCLA pre-COVID-19: Moderate	−0.062	0.940	0.505	0.897	−1.141	0.877
UCLA pre-COVID-19: Severe	0.914	2.494	0.505	0.040 *	0.022	2.000
Constant	−2.627	0.072	1.525	0.000	−4.412	−1.629

* *p* < 0.05, ** *p* < 0.01.

## Data Availability

Restrictions apply to the availability of the CLIC data. To request data access, readers should contact the co-author R.O.

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
