# Peer review of "Impact of COVID-19 Pandemic on Caregivers of People with an Intellectual Disability, in Comparison to Carers of Those with Other Disabilities and with Mental Health Issues: A Multicountry Study"

_ijerph, 2023, doi:10.3390/ijerph20043256_

Round 1

Reviewer 1 Report

The research tackles important topic of the impact of the COVID-19 pandemic on family career of people with an intellectual disability. I find the research topic interesting, important and timely, especially that there is a scarcity of previous work on the topic. Moreover, the unquestionable strength of the data comes from an international online survey conducted across 100 countries. Finally, the research itself was designed and described clearly. However, while I believe that this research fills the gap in the literature and may be of interest to the readers of the Journal there are some major issues that have to be revised before it could be published:

1. Although the methodological part is resented clearly, in my opinion it would be worth to work a bit on its structure. For better reception, it is worth dividing the content of the methodological part according to the following scheme: study design, participants and setting, research tools, data collection, ethical issues, data analysis. In its present form, some of these aspects are mixed.

2. In the Introduction some more information on the CLIC study should be given.

3. I am a bit surprised that no information on such important sociodemographic factors as education, family status and religion/spirituality health was given. Meanwhile, research show that they also influence caregivers’ burden. 

4. Although the Authors declare that the online survey was conducted among participants from 100 countries they provide no information on participants’ nationality/ethnic background. Meanwhile socio-cultural context is also very important and could have influenced caregivers’ experiences. Especially that while the subtitle of this manuscript refers to “A multicountry study” no information about participants from each country was given

5. As the paper if focused on the caregivers of people with an intellectual disability what was the rationale for including the carers of people with physical disability and mental illness? Either the title should be changed or both these groups should be excluded.

6. Lines 229, 261, 266, 278: Error! Reference source not found ???

7. Apart from highlighting the main conclusions, the final paragraph should provide indication of the direction future research should take. I suggest to discuss and give a more critical judgement on possible application of the results of the study. The Authors could also reflect more on the policy implications of their research: what solutions should be implemented in order to overcome the problem discussed in the manuscript. Thus, the paper would benefit from adding some recommendations suggesting possible guidelines for policymakers.

Author Response

Thank you for reviewing our article.  We have acted upon your comments and have made the changes requested. Please find attached our responses and the amendments made to the paper.

Yours sincerely

The Authors

Reviewer 1

The research tackles important topic of the impact of the COVID-19 pandemic on family career of people with an intellectual disability. I find the research topic interesting, important and timely, especially that there is a scarcity of previous work on the topic. Moreover, the unquestionable strength of the data comes from an international online survey conducted across 100 countries. Finally, the research itself was designed and described clearlyHowever, while I believe that this research fills the gap in the literature and may be of interest to the readers of the Journal there are some major issues that have to be revised before it could be published:

  1. Although the methodological part is resented clearly, in my opinion it would be worth to work a bit on its structure. For better reception, it is worth dividing the content of the methodological part according to the following scheme: study design, participants and setting, research tools, data collection, ethical issues, data analysis. In its present form, some of these aspects are mixed.

Reply: We have restructured the methods section to study design, participants, research tools, ethical approval and analysis

  1. In the Introduction some more information on the CLIC study should be given.

Reply: We have added the following paragraph to summarise the findings from the CLIC study

International data from the Coping with loneliness, isolation and caregiver burden COVID-19 (CLIC) study which received responses from over 100 countries across 10 languages reported significant rises in severe loneliness and isolation among carers who were are ready vulnerable with mental health or financial difficulties(O’Sullivan et al., 2021)

  1. I am a bit surprised that no information on such important sociodemographic factors as education, family status and religion/spirituality health was given. Meanwhile, research show that they also influence caregivers’ burden. 

We have added education, marital status and religion to table 1 and amended the text accordingly.

In each group the majority of participants were married (70%), with more people caring for those with dementia (SR=6.3) being married and fewer than expected caring for those with mental health difficulties (SR=-5.4). Religion was equally important across the groups (30.7%) and there were no significant differences in the amount of third level education received (71.6%).

  1. Although the Authors declare that the online survey was conducted among participants from 100 countries they provide no information on participants’ nationality/ethnic background. Meanwhile socio-cultural context is also very important and could have influenced caregivers’ experiences. Especially that while the subtitle of this manuscript refers to “A multicountry study” no information about participants from each country was given

Reply:  We have added information on the regions from where participants originated.

Participants were from North America (45.2%) United Kingdom and Ireland (19.4%), Europe (11.1%), Asia (10.4%) and others (13.9%).

  1. As the paper if focused on the caregivers of people with an intellectual disability what was the rationale for including the carers of people with physical disability and mental illness? Either the title should be changed or both these groups should be excluded.

Reply: We have amended the title to read:

Impact of COVID-19 pandemic on caregivers of people with an intellectual disability, in comparison to carers of those with other disabilities and with mental health issues: A multicountry study

  1. Lines 229, 261, 266, 278: Error! Reference source not found ???

Reply:  These have been sorted as were links to tables numbers that had become broken in the final preparations.

  1. Apart from highlighting the main conclusions, the final paragraph should provide indication of the direction future research should take. I suggest to discuss and give a more critical judgement on possible application of the results of the study. The Authors could also reflect more on the policy implications of their research: what solutions should be implemented in order to overcome the problem discussed in the manuscript. Thus, the paper would benefit from adding some recommendations suggesting possible guidelines for policymakers

Reply: We have added a recommendation for future research and modified a sentence to become a policy recommendation.

Future research should consider the burden experienced by vulnerable carers of people with an intellectual disability.

Policy changes may be needed to ensure that services are not locked down in future and given the scope for adaptations to meet the needs of carers.

Reviewer 2 Report

This is an interesting study dealing with a timely topic. However, it needs a lot of work to improve its quality, particularly regarding their study design and transparency. 

-Abstract should provide data on your study design and results. 

-Please, for the sake of clarity, provide a flowchart to follow-up how did you get your final sample. Is there any chance for selection bias? 

-Were there any dropouts?

-Provide number of ethical approval. 

-My main concern is related to the statistical method used. An adjusted regression would have been more convenient since you could have controlled confounding bias. The current estimates could be easily biased. 

-Format of acknowledgements should be corrected. 

Author Response

Thank you for reviewing our article.  We have acted upon your comments and have made the changes requested. Please find attached our responses and the amendments made to the paper.

Yours sincerely

The Authors

Reviewer 2

This is an interesting study dealing with a timely topic. However, it needs a lot of work to improve its quality, particularly regarding their study design and transparency. 

  1. Abstract should provide data on your study design and results. 

Reply: We have rewritten the abstract to provide data and increased detail on the design.

Carers supporting people with an intellectual disability often rely on others, to manage the burden of care. This research aims to compare the differences between carer groups and understand the predictors of loneliness changes and burden for carers of people with an intellectual disability. Data from the international CLIC study were analysed. In total 3,516 carers responded from four groups; people who care for those with Mental health difficulties (n=491), dementia (n=1888), physical disabilities (n=1147) and Intellectual disabilities (n=404). Cross tabulation and chi squared were used to compare group compositions and binary logistic regression to model predictors within the intellectual disability group. 65% of those caring for people with an intellectual disability experienced increased burden and 35% of carers of people with an intellectual disability and another condition experienced more severe loneliness. Becoming severely lonely was predicted by feeling burdened by caring (AOR,15.89) and worsening mental health (AOR,2.13)   Feeling burden was predicted by being aged between 35 and 44 (AOR, 4.24), poor mental health (AOR,3.51), and feelings of severe loneliness prior to the pandemic (AOR, 2.45).  These findings demonstrate that those who were already struggling with caring experienced the greatest difficulties during the COVID-19 lockdowns.

  1. Please, for the sake of clarity, provide a flowchart to follow-up how did you get your final sample. Is there any chance for selection bias? 

Reply:  We have added the following to the limitations section to ensure all readers are aware that there is potential that respondents were not fully representative:

The sample given the methods used cannot be considered representative. Also, individuals who completed the online questionnaire were not all in the same moment of lockdowns. There is potential for sample bias, for example it may be the voice of those who were negatively affected by the lockdowns was overrepresented. 

  1. Were there any dropouts?

Reply: This was an international online study.  If participants dropped out during the data collection process then their data would not have been recorded.  It is not possible to establish how many people started and did not complete the study.

  1. Provide number of ethical approval. 

There was no specific ethics approval number however the date on the ethics approval form reads: UU 15.05.20

  1. My main concern is related to the statistical method used. An adjusted regression would have been more convenient since you could have controlled confounding bias. The current estimates could be easily biased. 

Reply: We did use an adjusted odds ratios for these reasons.

  1. Format of acknowledgements should be corrected. 

Reply: Thank you for pointing this out we have adjusted the formatting to match with the rest of the paper.

Round 2

Reviewer 1 Report

The Authors have clarified several issues raised in the review and this revised manuscript is now more consistent owing to their corrections and additional arguments. At the same time, whole I appreciate this effort I still believe that some other issues require further explanation:

I believe that the full name of the CLIC study should be written with capital letters? Additionally, the Authors should describe its aim (was it to examine the psychological impact of the COVID-19 pandemic through validated self‐report measures of loneliness and social isolation?)

Still no information on countries included in the study was given. Instead, the Authors, enumerate several continents (i.e. North America, Europe, Asia, Latin America, Africa and Oceania). At the same time they do mention the United Kingdom and Ireland, why?

Additionally, there is no information on participants’ nationality/ethnic background.

Finally, at least in the Discussion section I would expect at least some comparison between countries as many research show that culture, including religion, is important factor that influences caregivers experiences. Moreover, as different countries implemented different public health interventions and control measures I would recommend refer to these differences in the Discussion section.

There is some problem with Table 1.

The first paragraph of the Conclusions section describes study limitations and should be removed.

More importantly, in its current form the Conclusions are very vague and should be rewritten with some recommendations addressing the main findings from  the study. Simple statement that: “Policy changes may be needed to ensure that services are not be locked down in future but they should be adapted and given the scope for adaptations to meet the needs of carer” is not enough.

Reviewer 2 Report

The article has critical methodology flaws and the authors did not successfully amend the manucript and respond in accordance with the raised points. 
